# Declining Epstein-Barr Virus Antibody Prevalence in College Freshmen Strengthens the Rationale for a Prophylactic EBV Vaccine

**DOI:** 10.3390/vaccines10091399

**Published:** 2022-08-26

**Authors:** Henry H. Balfour, Madeline R. Meirhaeghe, Arianna L. Stancari, Jennifer M. Geris, Lawrence M. Condon, Laurel E. Cederberg

**Affiliations:** 1Department of Laboratory Medicine and Pathology, University of Minnesota Medical School, Minneapolis, MN 55455, USA; 2Department of Pediatrics, University of Minnesota Medical School, Minneapolis, MN 55455, USA; 3Health Partners, St. Paul, MN 55108, USA

**Keywords:** EBV infection, infectious mononucleosis, EBV antibody prevalence, EBV vaccine

## Abstract

Background: To better understand the epidemiology of primary Epstein-Barr virus (EBV) infection and to identify EBV-naïve candidates eligible to receive a prophylactic EBV vaccine, we screened freshmen from the University of Minnesota Class of 2025 for circulating EBV antibody, which is indicative of previous infection. This permitted us to compare their EBV antibody prevalence with that of 4 other freshman classes (Classes of 2010, 2011, 2016, 2021) that have been previously published. Methods: Freshman students were recruited during screening sessions in the residence halls. Venous blood was collected and the serum fraction tested for IgG antibody against EBV viral capsid antigen (VCA IgG) using commercial enzyme immunoassays. Results: All classes combined, 1196 participants were tested (female, 677; male, 513; did not identify gender, 6) who were 18–23 years old (median, 18; mean, 18.37). The EBV VCA IgG antibody prevalence was 58% (689/1196) and was higher in women than men. The EBV antibody prevalence of 64% (170/267) in the 2010 freshman class versus 52% (78/150) in the Class of 2025 was statistically significantly different (*p* = 0.0223, Fisher exact test).” Conclusions: Sufficient participants are available for a prophylactic vaccine trial. Antibody prevalence decreased over 15 years from 64% to 52%. If this trend continues, the number of EBV-naïve adolescents and young adults who are in the age group most susceptible to infectious mononucleosis will increase, strengthening the rationale to develop an effective prophylactic EBV vaccine.

## 1. Introduction

Epstein-Barr virus (EBV) is a ubiquitous human virus, infecting at least 90% of adults worldwide [1]. Diseases due to or spurred by EBV include infectious mononucleosis, endemic Burkitt lymphoma, cancers after transplantation, Hodgkin lymphoma, nasopharyngeal carcinoma, and autoimmune conditions, especially multiple sclerosis [2,3]. Our hypothesis is that a prophylactic EBV vaccine has the potential to reduce the incidence and/or the severity of all these diseases [4].

Prophylactic EBV vaccines are under development. One has shown efficacy in a phase 2 clinical trial [5] and at least two others are currently in clinical trials. To identify EBV-naïve candidates who would be eligible to receive a prophylactic EBV vaccine, we performed a series of studies that screened college freshmen for the presence of circulating EBV antibody. This also permitted us to compare the EBV infection status of five freshman classes over a 15-year period and to assess the feasibility of conducting vaccine trials with EBV-naïve participants on a university campus.

## 2. Materials and Methods

### 2.1. Study Population

Participants from the classes of 2010, 2011, 2016, 2017, and 2025 were included. Studies involving the first 4 classes have previously been published [6,7,8]. The method of recruitment was the same for all 5 classes. Freshman students were notified about the studies through emails and campus posters facilitated by the University of Minnesota Division of Housing and Residential Life. Intake sessions were conducted in the residence halls. After giving written informed consent, participants completed a brief demographic questionnaire that included age in years and gender, and donated approximately 10 mL of venous blood. They were informed of their results by email and offered a time to discuss them with the principal investigator (HHB). All of the studies were approved by the University of Minnesota Institutional Review Board. 

### 2.2. EBV Antibody Assay

Sera were separated from the clot and stored at −80 °C until they were tested for IgG antibody against EBV viral capsid antigen (VCA) using commercially available enzyme immunoassay (EIA) kits from Diamedix (Miami, Florida) for the first 4 classes. Kits from Tecan (Männedorf, Switzerland) were used for the class of 2025 because Diamedix kits were not available. Results were expressed as the index value, which was the absorbance of the participant’s sample divided by the mean absorbance of 3 replicate dilutions of a weakly positive control supplied by the manufacturer. EIA results were classified according to their index value as negative, <0.90; equivocal, 0.90–1.09; or positive, >1.09. Participants whose EBV VCA IgG antibody indices were in the negative range were considered to be EBV-naïve. As 2 different kits were used to detect EBV VCA IgG antibody, a head-to-head comparison was performed on stored samples.

### 2.3. Statistical Analysis

Comparisons were analyzed by the Fisher exact test. (https://www.socscistatistics.com/tests/fisher/default2.aspx, accessed on 14 July 2022). 2-sided *p*-values less than <0.05 were considered statistically significant.

## 3. Results

### 3.1. EBV VCA IgG Antibody Kit Comparison

The Diamedix and Tecan EBV VCA IgG antibody kits were qualitatively identical. Results were concurrent for 19 of 20 samples tested in parallel. The one discordant sample was negative for EBV VCA IgG antibody by Diamedix, but equivocal by Tecan.

### 3.2. EBV Antibody Prevalence

Samples were tested from 1196 students (female, 677; male, 513; did not identify gender, 6) who were 18–23 years old (median, 18; mean, 18.37). As shown in Table 1, the EBV VCA IgG antibody prevalence declined from 64% for the Class of 2010 to 52% for the Class of 2025, which was statistically significant (*p* = 0.0223, Fisher exact test). In all 5 classes tested, the EBV VCA IgG antibody prevalence was higher in women than men.

## 4. Discussion

Circulating antibodies of the IgG class against EBV VCA and EBV nuclear antigen (EBNA) indicate a previous EBV infection. In our first prospective study, all 66 participants who contracted a primary EBV infection developed VCA IgG antibodies, whereas 5 of 62 volunteers (8%) with a primary EBV infection who were followed for at least a year remained EBNA IgG antibody negative [6]. Therefore, we chose to test for VCA IgG instead of EBNA IgG to determine EBV serostatus.

As the recruitment strategies were identical for all classes and the ages of the students very similar, we believed that comparing the classes was reasonable. The prevalence of EBV VCA IgG antibody declined statistically significantly when students from the class of 2025 were compared with those from the classes of 2010, and the decrease started before the COVID-19 pandemic. A decrease in antibody prevalence has also been observed among Japanese children [9], among participants in the National Health and Nutrition Examination Surveys [10], and among patients in eastern France [11]. The reason for this is not clear although some have attributed it to a general improvement in hygiene. If the trend continues, the number of adolescents and young adults who are most susceptible to infectious mononucleosis will increase, emphasizing the need to develop effective prophylaxis and treatment strategies. This also could increase the number of adults who contract infectious mononucleosis over the age of 40 and are likely to have more severe disease than younger patients [12].

EBV antibody prevalence was consistently higher in women than men. This could be because females engage in deep kissing at a younger age than males.

We explored feasibility of multicenter EBV vaccine trials on college campuses by collaborating on a cross-sectional EBV antibody study of University of Iowa freshmen from the Class of 2022 [13]. Of 198 participants tested, 87 (44%) were EBV-naïve. This is identical to what was found in University of Minnesota freshmen from the Class of 2021. Of 221 participants tested, 98 (44%) were EBV-naïve.

A prophylactic EBV vaccine has the potential to reduce the incidence and/or the severity of infectious mononucleosis, which is a scourge on college campuses. The cross-sectional studies at the University of Minnesota and the University of Iowa support the feasibility of conducting vaccine trials with EBV-naïve participants at more than one college campus.

## Figures and Tables

**Table 1 vaccines-10-01399-t001:** Epstein-Barr Virus Antibody Prevalence in University of Minnesota Freshmen.

Study Population	Class of 2010	Class of 2011	Class of 2016	Class of 2021	Class of 2025
Month/Year Tested	September/October 2006	September 2007	September 2012	September/October 2017	November/December 2021 April 2022
No. of Participants	267	279	279	221	150
Age Range in Years (median, mean)	18–22 (18.6, 18.6)	18–23 (18.6, 18.6)	18–21 (18.6, 18.6)	18–19 (18.7, 18.7)	18–20 (18.0, 18.4)
Antibody Positive Male	59/99 (60%)	70/121 (58%)	78/167 (47%)	47/85 (55%)	21/41 (51%)
Antibody Positive Female	111/168 (66%)	104/158 (66%)	66/112 (59%)	75/135 (56%)	55/104 (53%)
Antibody Positive Male & Female	170/267 (64%)	174/279 (62%)	144/279 (52%)	123/221 (56%) *	78/150 (52%) **

* One participant did not identify their gender. ** Five participants did not identify their gender.

## Data Availability

The data presented in this study are available on request from the corresponding author. The data are not publicly available due to privacy considerations.

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
