# Peer review of "Declining Epstein-Barr Virus Antibody Prevalence in College Freshmen Strengthens the Rationale for a Prophylactic EBV Vaccine"

_vaccines, 2022, doi:10.3390/vaccines10091399_

Round 1

Reviewer 1 Report

The authors have previously published EBV-serostatus of college freshmen classes from 2010 to 2017 and now add data from the 2025 class. EBV positive serostatus is gradually and significantly declining during this period, and the authors conclude this strengthens the rationale for an EBV vaccine in this age group.

This is a small but well-performed study adding significant value to their previous articles, and I just have some minor issues (line number where appropriate):

-23: "Overall" could perhaps read "All classes combined", I first read it as referring to the 2025 class only

-26-27: The number of participants in the class of 2010 and 2025 should be added in brackets (n=....)

-27 and 84: These two p-values are not the same, although they refer to the same analyses?

-55: How was the serum prepared? Was it frozen before bulk analyses?

-EBV seropositivity would be even lower in younger age groups, would it not be reasonable to target pre-adolescents for vaccine (even if a trial should start with adults)? 

-The (presumably) delayed exposure to EBV might have significant clinical consequences, it would be interesting to have a comment from the authors also on this.

Author Response

Reviewer 1

This is a small but well-performed study adding significant value to their previous articles, and I just have some minor issues (line number where appropriate):

1.-23: "Overall" could perhaps read "All classes combined", I first read it as referring to the 2025 class only

We agree and have changed that sentence to: “All classes combined, 1,196 participants were tested…”

2.-26-27: The number of participants in the class of 2010 and 2025 should be added in brackets (n=....)

We agree and have changed the sentence to: “The EBV antibody prevalence of 64% (170/267 ) in the 2010 freshman class versus 52% (78/150) in the Class of 2025 was statistically significantly different (P = 0.0223, Fisher exact test).”

  1. -27 and 84: These two p-values are not the same, although they refer to the same analyses?

Excellent point. These actually were different analyses but to avoid confusion we have compared in both places the Class of 2010 versus the Class of 2025 instead of the Classes of 2010-11 versus the Class of 2025. Please see point 2 above.

  1. -55: How was the serum prepared? Was it frozen before bulk analyses?

We have added: “Sera were separated from the clot and stored at -80c until they were tested...”

  1. -EBV seropositivity would be even lower in younger age groups, would it not be reasonable to target pre-adolescents for vaccine (even if a trial should start with adults)? 

Great point and it agrees with our plans to ultimately make this a pediatric vaccine.

6.-The (presumably) delayed exposure to EBV might have significant clinical consequences, it would be interesting to have a comment from the authors also on this.

There are observations to support the notion that primary EBV infection becomes more severe with increasing age. In response to this reviewer’s comment, we have added a sentence and reference beginning on line 113: “This also could increase the number of adults who contract infectious mononucleosis over the age of 40 and are likely to have more severe disease than younger patients [7].”

Reviewer 2 Report

The paper is very short and is not suitable for publication in that form. The authors only assessed the seroprevalence of EBVCA IgG among students of Minnesota University. They observed a decrease in the incidence of antibodies and suggest that it will increase the number of adolescents and young adults who are most  susceptible to infectious mononucleosis. 

My comments are following: 1.     In the introduction, it should be explained why, according to the authors, such an analysis is important. Is there a high incidence of mononucleosis? Is there a prophylactic vaccine against EBV available. If not, how far is the research work advanced. 2.     quantitative tests are now commercially available. It would be interesting if the titer of these antibodies is also declined. 3.     The discussion could explain why the prevalence of antibodies is decreasing. 4.     In my opinion, quantitative research should be carried out, and then correct this version of the article.

Author Response

Reviewer 2

This reviewer checked the box, “Extensive editing of English language and style required.”

We humbly disagree. English is the native language of all the authors.

My comments are following: 

  1. In the introduction, it should be explained why, according to the authors, such an analysis is important. Is there a high incidence of mononucleosis? Is there a prophylactic vaccine against EBV available. If not, how far is the research work advanced. 

The primary goal of this research was to learn if there were sufficient numbers of EBV-naïve students to make a clinical trial of a prophylactic EBV vaccine feasible. I believe we say that but in accordance with this comment we have added a sentence beginning on line 41: “Prophylactic EBV vaccines are under development and at least 2 are in clinical trials.”

  1. Quantitative tests are now commercially available. It would be interesting if the titer of these antibodies is also declined. 

The purpose of this study was a gauge how many EBV-naïve students were available to make a clinical trial of a prophylactic EBV vaccine feasible. The quantity of antibody in seropositive students was not relevant. Also, the quantity of antibody will fluctuate with reactivation as well as external exposure to the virus.

  1. The discussion could explain why the prevalence of antibodies is decreasing. 

We do, beginning on line 107: “A decrease in antibody prevalence has also been observed among Japanese children [5], and among participants in the National Health and Nutrition Examination Surveys [6]. The reason for this is not clear although some have attributed it to a general improvement in hygiene.”              

  1. In my opinion, quantitative research should be carried out, and then correct this version of the article.

Please see our response to Comment 2 above.

Reviewer 3 Report

The present study aimed to detect differences in anti-EBV antibodies in the serum of freshmen from classes between 2010 and 2025 (so 2006-2022).

The study is per se interesting. Personally, I'm even in favor of anti EBV vaccination, due to its involvement in cancer.

However, I don't see how data can support the conclusions. Similar percentages are observed since 2012 (52%) and overall, a minus 10% between 2006 and 2022, despite statistically significant, is not strong enough (given the intermediate values) to draw clinically relevant conclusions.

Author Response

Reviewer 3

I don't see how data can support the conclusions. Similar percentages are observed since 2012 (52%) and overall, a minus 10% between 2006 and 2022, despite statistically significant, is not strong enough (given the intermediate values) to draw clinically relevant conclusions.

There will always be sampling variations in clinical research, but the bottom line, supported by the results of the National Health and Nutrition Examination Surveys and Japanese studies cited in our manuscript, is that EBV seroprevalence is decreasing. Our data show a decrease of 12% from 2006 to 2021-22 (P = 0.0223 ). This difference is clinically relevant to recruitment of volunteers for prophylactic EBV vaccine trials.

Round 2

Reviewer 2 Report

Authors revised the manuscript according all comments.

This paper may be accepted.

Author Response

Reviewer comments: Authors revised the manuscript according all comments. This paper may be accepted.

Our response: We thank this reviewer for their time and attention to our manuscript.